# The Role of ctDNA for Diagnosis and Histological Prediction in Early Stage Non-Small-Cell Lung Cancer: A Narrative Review

**DOI:** 10.3390/diagnostics15070904

**Published:** 2025-04-01

**Authors:** Carolina Sassorossi, Jessica Evangelista, Alessio Stefani, Marco Chiappetta, Antonella Martino, Annalisa Campanella, Elisa De Paolis, Dania Nachira, Marzia Del Re, Francesco Guerrera, Luca Boldrini, Andrea Urbani, Stefano Margaritora, Angelo Minucci, Emilio Bria, Filippo Lococo

**Affiliations:** 1Thoracic Surgery, Fondazione Policlinico Universitario A. Gemelli Istituto di Ricovero e Cura a Carattere Scientifico (IRCCS), 00168 Rome, Italy; marcokiaps@hotmail.it (M.C.); annalisa.campanella@guest.policlinicogemelli.it (A.C.); dania.nachira@policlinicogemelli.it (D.N.); stefano.margaritora@policlinicogemelli.it (S.M.); filippo.lococo@policlinicogemelli.it (F.L.); 2Medical Oncology, Department of Translational Medicine and Surgery, Università Cattolica del Sacro Cuore, 00168 Rome, Italy; alessio.stefani@unicatt.it (A.S.); emilio.bria@policlinicogemelli.it (E.B.); 3Thoracic Surgery Unit, University “Magna Graecia”, 88100 Catanzaro, Italy; 4Radiotherapy Unit, A. Gemelli University Hospital Foundation IRCCS, 00168 Rome, Italy; antonella.martino@policlinicogemelli.it; 5Departmental Unit of Molecular and Genomic Diagnostics, Genomics Research Core Facility, Gemelli Science and Technology Park (GSTeP), Fondazione Policlinico Universitario A. Gemelli IRCCS, 00168 Rome, Italy; elisa.depaolis@policlinicogemelli.it (E.D.P.); angelo.minucci@policlinicogemelli.it (A.M.); 6Clinical Chemistry, Biochemistry and Molecular Biology Operations (UOC), Fondazione Policlinico Universitario A. Gemelli IRCCS, 00168 Rome, Italy; andrea.urbani@policlinicogemelli.it; 7Thoracic Surgery Unit, Università Cattolica del Sacro Cuore, 00168 Rome, Italy; 8Department of Faculty Medicine, Saint Camillus International University of Medical and Health Sciences, 00131 Rome, Italy; marzia.delre@gmail.com; 9Department of Cardio-Thoracic and Vascular Surgery, Azienda Ospedaliera-Universitaria Città Della Salute e Della Scienza di Torino, 10126 Torino, Italy; francesco.guerrera@unito.it; 10Department of Surgical Sciences, University of Torino, 10126 Torino, Italy; 11Department of Radiology, Radiotherapy and Hematology, Fondazione Policlinico Universitario A. Gemelli IRCCS, 00168 Rome, Italy; luca.boldrini@policlinicogemelli.it; 12Department of Basic Biotechnological Sciences, Intensivological and Perioperative Clinics, Catholic University of Sacred Heart, 00168 Rome, Italy; 13UOC Oncologia Medica, Isola Tiberina Gemelli Isola, 00186 Rome, Italy

**Keywords:** non-small-cell lung cancer, early stage diagnosis, circulating tumor DNA

## Abstract

**Background:** Circulating tumor DNA (ctDNA) may be released from neoplastic cells into biological fluids through apoptosis, necrosis, or active release. In patients with non-small-cell lung cancer (NSCLC), ctDNA analysis is being introduced in clinical practice only for advanced disease management. Nevertheless, an interesting and promising field of application is the analysis of ctDNA in the management of early stage non-small-cell lung cancer, both for evaluation before treatment, such as diagnosis and screening, and for prediction of histology or pathological features. **Methods:** A thorough review of the literature published between 2000 and 2024 was performed on PubMed, utilizing the advanced search feature to narrow down titles and abstracts containing the following keywords: ctDNA, early stage, and NSCLC. A total of 20 studies that met all inclusion criteria were chosen for this review. **Results:** In this review, we summarize the increasing evidence suggesting that ctDNA has potential clinical applications in the management of patients with early stage NSCLC. ctDNA levels in early stage cancers are very low, posing many technical challenges in improving the detection rate and sensitivity, especially in clinical practice, if it is to be implemented for early detection. Presently, the main limitation of ctDNA experimental and clinical studies, especially in early stage settings, is the lack of definitive standardization and consensus regarding methodology, the absence of systematically validated analyses, and the lack of adoption of sensitive approaches. **Conclusions:** Possible applications of this analyte open up new fields of diagnosis, treatment, and follow up, which are less invasive and more precise than other approaches currently in use, especially in early stage NSCLC patients.

## 1. Introduction

Tumor molecular characteristics identification is key for the implementation of targeted antineoplastic treatments with a higher chance of success. Tumor-derived materials that are detectable in blood and in other biological fluids may be useful for cancer gene analysis. The process of obtaining genetic material from biological fluids is known as liquid biopsy. This approach overcomes the limitations of tissue analysis, such as the potential difficulties in organizing a surgical biopsy. It is repeatable over time and is furthermore well tolerated by patients, as it is non-invasive. Usually, a liquid biopsy is just a blood sample [1]. The concept of liquid biopsy emerged 10 years ago for the identification of circulating tumor cells. Its applications rapidly expanded to other tumor-derived products such as circulating tumor DNA (ctDNA), circulating cell-free RNA, extracellular vesicles, or tumor-educated platelets [2]. Concerning ctDNA, its role as a prognostic marker has already been documented in many kinds of tumor such as breast, prostate, lung, and colorectal cancers [3,4,5]. In 2022, the European Society of Medical Oncology (ESMO) published a recommendation manuscript focused on ctDNA detected in plasma as a liquid biopsy analyte, and highlighted that ctDNA has sufficiently strong evidence to be used routinely in clinical practice to genotype cancers for direct molecularly targeted therapies [6]. For this reason, the implementation of ctDNA from liquid biopsies during tumor diagnosis work up is starting to be recommended by international oncology societies with the use of next-generation sequencing (NGS) for tumor molecular assessment. ctDNA is generally more abundant in patients with advanced disease than in ones with early stage lung cancer [7]. Different strategies are now in use to detect ctDNA in early stage disease. In particular, methylation and fragmentomic profiles are emerging as promising biomarkers for the identification of ctDNA in the early stages of tumor development [8,9].

An interesting and promising field of application is the analysis of ctDNA in the management of early stage non-small-cell lung cancer (NSCLC), both for evaluation before treatment and for prediction of histology or pathological features [7]. The possible application of this analyte opens new fields of diagnosis, treatment, and follow up, which are less invasive and much more precise than other approaches currently in use, especially in early stage NSCLC patients.

The aim of this narrative review is to highlight the most relevant applications of ctDNA, starting from technical aspects and moving through its diagnostic and predictive role in the histology of early stage lung cancer. A focus will also be made on the limitations of this approach and also on future perspectives.

## 2. Methods

A comprehensive search of the literature published from 2000 to 2024 was conducted on PubMed. Titles and abstracts were filtered using the advanced search tool and the following keywords: ctDNA, early stage, and NSCLC.

Year of publication: Any publication date starting from 1 January 2000 to 30 November 2024 was eligible. Language: Only studies with their full text in the English language were included.

Type of study: Only peer-reviewed publications reporting primary data were eligible. Therefore, reviews, editorials, letters, and other forms of secondary expert opinion were excluded at the screening stage. Only full manuscripts were eligible, excluding conference abstracts and proceedings. No constraints were imposed based on the level of evidence.

We included all original studies describing methods of identification of ctDNA, baseline ctDNA analysis for diagnosis and screening, histopathological features, and staging of early stage lung cancer.

The following data were extracted onto a Microsoft Excel spreadsheet: author, period, country, kind of study, and finality of ctDNA use. The final manuscript was shared with the different principal investigators of eligible studies (co-authors of the present study) and the final manuscript was approved by all co-authors.

The exclusion criteria included the following: the use of ctDNA analysis for other kinds of tumors and for advanced lung cancer and the use of other blood features for lung cancer studies, case reports, case series, and reviews (The search strategy is summarized in Table 1).

## 3. Literature Research Outcome

In the comprehensive literature search on PubMed, 73 articles were found. By reviewing titles and abstracts, articles formatted as reviews, editorials, letters, commentaries or case reports, three duplicates, non-English language articles were excluded. Sixty eligible studies were selected and retrieved in their full-text version; no additional study was found by cross-reference.

Forty full-text reports were excluded for the following reasons: being focused on ctDNA levels after radiotherapy or surgery and for the prediction of lung cancer recurrence, being written in a language different from English, and having a study group with less than 10 cases. Finally, 20 studies met all inclusion criteria and were selected for the present review.

### 3.1. Methods of Identification of ctDNA

In patients with cancer, cell-free DNA (cfDNA) and the fraction derived from tumor cells, namely circulating tumor DNA (ctDNA), are shed into blood and other body fluids [10]. The specific clinical need informs the choice of the laboratory analytical approach, with great variability in terms of assay type and chemistry, sequencing gene panel size and content, and so on. Commonly, ctDNA mutations (such as chromosomal rearrangements, copy number alterations, and epigenetic changes as methylation patterns), detected for both pre-therapeutic and post-therapy purposes, are identified using two types of approaches: (i) targeted analysis for the evaluation of a few known molecular alterations; (ii) untargeted analysis for a broad evaluation of ctDNA aberrations [11]. Methods of ctDNA identification are summarized in Table 2. Targeted analysis mainly includes quantitative real-time polymerase chain reactions (qPCRs) [12], digital PCRs (dPCRs) [13], beads, emulsion, amplification, and magnetics (BEAMing) technologies [14], and next-generation sequencing (NGS)-based assays such as the cancer personalized profiling by deep sequencing (CAPP-Seq) [15]. The adoption of such assays is commonly tumor-informed, according to the availability of a previous genotyped result obtained from tumor tissue biopsy. Several literature studies have established the clinical utility and feasibility of targeted ctDNA mutation analysis in early stage lung cancer patients, with a certain degree of concordance rate with matched tumor tissue genotyping. Guo and colleagues applied a 50-gene targeted sequencing approach to identify and monitor molecular changes in ctDNA mutation frequencies in 30 early stage NSCLC patients (I–II) previously genotyped on tissue samples, comparing results with those obtained in advanced disease. The authors reported a higher tDNA/ctDNA concordance rate (80%) in early stage patients than in late-stage patients, with a lower specificity explained by the increased plasma ctDNA levels in advanced settings. The analysis of pre- and post-surgery time points revealed a major decrease in mutation frequency in stage I as compared with more advanced settings [16]. Chen and colleagues evaluated 76 stages of I–IIIA lung cancer patients by using a 50-gene-based NGS panel, obtaining a concordance rate that mainly depended on the tumor stage (stage I, 57.9%; stage II, 66.7%; stage III, 90%), similar to other studies [17,18]. Overall, sequencing strategies indicated that ctDNA mutational analysis in early stage lung cancer is feasible even if it requires more sensitive and optimized methods [19]. Recently, a targeted NGS panel named ultra-high-sensitivity lung version 1 (ULV1) was developed to detect hotspot mutations in a low fraction of cfDNA of NSCLC patients, demonstrating a good detection sensitivity and a high degree of analytical sensitivity and specificity [20]. High-throughput untargeted assays include whole-exome sequencing (WES), whole-genome sequencing (WGS), and mass-spectrometry-based assays [21]. These approaches allow a widespread and tumor-agnostic analysis of cfDNA but they might not always be feasible with a limited amount of ctDNA (as is often the case in early stage settings), making their application limited [10]. In the context of screening and early detection, other approaches based on DNA fragmentation [22] and methylation analysis were also described in the literature, although not implemented in clinical practice. Tran and colleagues used a hybrid approach based on a targeted NGS panel and a methylation pattern analysis, proving that the ctDNA clearance in early stage NSCLC allows risk stratification after treatments [23]. Hong and colleagues adopted the Grail multicancer early detection (MCED) technology, developed to differentiate the cell type or the tissue origin of cfDNA according to the methylation pattern in asymptomatic subjects, to support the prognostic utility of the ctDNA detection rate at a 2-year time point in pre-surgical stage I lung adenocarcinoma [24]. The same approach was adopted in the study of Bossè et al. [25], which proved that the prognostic value of the methylation Grail MCED test assessed in the preoperative step did not associate with disease relapse within 5 years. The potential role at 2 years post-surgery is retained for the stage I lung adenocarcinoma histology subtype [25]. The critical need of sensitive methodology adoption in early stage lung cancer is mainly related to the low ctDNA detection rates in this setting. Typically, the ctDNA fraction in a total cfDNA sample is very variable, ranging from 0.1% to 90% depending on cancer type and stage [26]. In patients with lung cancer, ctDNA is detectable in both early and late stages [27,28]. Specifically, fractions of ctDNA typically have <0.1% variant allele frequency (VAF) in stage I lung cancer patients [28,29], resulting in two-to-three mutant allele copies in 10 ng of cfDNA. General practical recommendations in the challenging scenario of early stage lung cancer liquid biopsy include the use of a high plasma input in the cfDNA isolation and a high cfDNA input in the amplification and/or sequencing analytical steps to ensure a low minor allele frequency (MAF) variant detection [20,30]. Not all the cfDNA consists of ctDNA and not all the ctDNA molecules harbor the tumor aberration at a specific locus, due to zygosity status and/or intratumor heterogeneity [31]. Thus, only a proportion of ctDNA fragments contain cancer-associated alleles as somatic small variants, abnormal methylation, and/or copy number alterations. This proportion is defined as the circulating tumor allele fraction (cTAF) [32]. The cTAF is used as a measure of ctDNA and its estimation relies on tumor-derived small-variant detection obtained through several approaches (deep sequencing, dPCR, qPCR) [32], chromosomal copy number aberrations using DNA shallow WGS (i.e., the ichorCNA test) [33,34], and methylation patterns (i.e., the tumor methylated fraction, TMeF) adopting targeted sequencing, whole-genome bisulfite sequencing (WGBS), and qPCR [32,35]. At present, the main limitation of ctDNA experimental and clinical studies, especially in early stage settings, is the lack of definitive standardization and consensus about methodological aspects, the absence of systematically validated analyses, and the lack of a sensitive approach adoption. Recently, the Association for Molecular Pathology Clinical Practice Committee’s Liquid Biopsy Working Group (LBxWG), including the American Society of Clinical Oncology (ASCO) and the College of American Pathologists (CAP) organizations, developed a set of best practice consensus recommendations for clinical ctDNA assays that need to be incorporated into clinical diagnostic routines [36].

### 3.2. Baseline ctDNA, Lung Cancer Screening, and Diagnosis of Suspected Malignancy

Diagnosis of non-small-cell lung cancer at an early stage is associated with favorable survival outcomes. According to the VIII edition of the TNM staging system of lung cancer by the International Association for the Study of Lung Cancer (IASLC) Lung Cancer Staging Project [37], clinical stage IA1 NSCLC (T1aN0M0) is associated with a 5-year overall survival of 92%, compared with 0% for clinical stage IVB NSCLC (any T, any N, M1c) [38]. Though invasive approaches (e.g., bronchoscopy, bronchoalveolar lavage, transthoracic needle aspiration) are available to confirm or rule out the diagnosis of lung cancer in patients with indeterminate lung nodules, consequent complications may occur including hemorrhage, infection, and pneumothorax. Accordingly, thoracic surgeons tend to select symptomatic patients with pulmonary nodules for invasive tests, which also leads to asymptomatic but true lung cancer patients missing the opportunity of early detection [39]. Therefore, there is an ongoing quest to identify the most effective way to predict the malignancy of a lung nodule. A few studies have reported the use of blood biomarkers for the diagnosis of lung cancer: they are used to distinguish normal tissue from malignancy. Welliver and coworkers [40] described in their analysis that lung cancer cells release DNA fragments into circulation, and these fragments can be found in the cell-free fraction of blood together with DNA fragments from normal cells (ctDNA). It has been hypothesized that ctDNA is passively released into the bloodstream from dying tumor cells, a process related to tumor burden, tumor growth, and anti-tumor therapy. Nevertheless, the identification of ctDNA during the early stages of cancer implies the challenge of the detection of the biomarkers, as levels of ctDNA are lower for initial tumors. Abbosh and colleagues [38] highlighted that patients with stage I NSCLC have the largest proportion of undetectable ctDNA using current NGS approaches. Also, Xu and coworkers [41] in 2022 considered the ctDNA level in early stage cancers to be very low, which posed many technical challenges in improving the detection rate and sensitivity, especially in clinical practice, if it is to be implemented in an early detection program. To solve these challenges, they developed an ultra-sensitive ctDNA MRD detection system in this study, namely the Personalized Analysis of Cancer (PEAC), to simultaneously detect up to 37 mutations, which account for 70–80% non-small-cell lung cancer (NSCLC) driver mutations from low plasma sample volumes. They concluded that the PEAC system developed can detect the majority of NSCLC driver mutations using 8–10 mL plasma samples, with the advantages of high detection sensitivity. This makes this system quite appropriate for early ctDNA detection in early stage NSCLC and thus to distinguish non-malignant lung nodules from real lung cancer. The question is also well analyzed by McKelvey and colleagues [42] in a retrospective study that included eight commercial ctDNA assay developers providing summary-level de-identified data for patients with non-small-cell lung cancer (NSCLC). For those samples with detected ctDNA, late-stage NSCLC samples generally appeared to have higher levels as compared to early stage samples.

In 2014, Newman and coworkers [18] explored whether ctDNA analysis could potentially be used for cancer screening and biopsy-free tumor genotyping. According to their experience, they implemented their cancer screening method for high specificity, and correctly classified 100% of patient plasma samples with ctDNA above fractional abundances of 0.4% with a false-positive rate of 0%. One of the most recent attempts to improve the early detection/screening approach comes from He and colleagues [39]. They validated a combined model using ctDNA methylation status, the largest nodule size measured by a low-dose CT scan, and patient age to better discriminate malignant from benign lung nodules. The proposed model proved to be superior to both low-dose CT-scan-only and methylation-only models in recognizing malignant lung nodules with an AUC of 0.89 (sensitivity: 91.21%) and showed high sensitivity with moderate specificity, suggesting its potential to improve the management of lung cancer screening. They proposed a novel cost-effective workflow for lung cancer screening in high-risk populations, which is as follows: CT scans to identify suspicious nodules, and subsequently a combined model that integrates the ctDNA-based methylation status of targeted genes, largest nodule size measured by LDCT, and age. Another important observation comes from Chen et al. [43], who showed that the detection of ctDNA mutations in peripheral blood is highly feasible for the early screening and diagnosis of lung cancer. Indeed, they observed that a low-dose CT scan is often overused, in non-smoking patients also, with a false-positive rate of 96% [44]. Significant advancements in plasma ctDNA detection are expected in the next 5–10 years. ctDNA is anticipated to be a key player in the early screening of NSCLC patients [45]. The results from the analyzed studies suggest that there may be room for ctDNA analysis in screening programs, as suggested by the high correspondence between ctDNA detection and lung cancer diagnosis on surgical specimens [18,42]. It is foreseeable that ctDNA detection will play a crucial role in guiding the comprehensive management of lung cancer patients in the future. The main results are summarized in Table 3.

### 3.3. Baseline ctDNA and Prediction of Histology, Pathological Features, and Staging

Lung adenocarcinoma presents with significantly lower ctDNA levels compared to squamous-cell carcinoma or other histotypes [24,43,46,47], with ctDNA detected within a range of about 13–40% of lung adenocarcinoma cases compared to 80% of squamous-cell carcinoma cases. This difference is more marked when the examination is focused on stage I only, with ctDNA detected in only about 13% of cases.

Interestingly, some studies had also evaluated the differences in ctDNA or cfDNA concentrations by comparing GGO/part-solid nodules with solid adenocarcinomas, with a significant difference reported by Hong et al. [24] in the ctDNA concentration, while Chen et al. [43] reported that patients with GGO-predominant tumors had a cfDNA concentration more than 10-times lower that in solid tumors.

Finally, Zhang et al. [46] explored the concordance between ctDNA and tissue DNA mutation, finding that the ratio was higher in squamous-cell carcinoma compared to adenocarcinoma, and developed a model to differentiate adenocarcinoma from squamous-cell carcinoma using an NGS panel of 546 genes. In detail, they selected 14 genes (TP53 SLIT2, NOTCH3, MTOR, LIFR, MRE11A, ARID2, ERCC3, KCNH2, CDC25, RB1, ALK, NFE2L2, and FBXW7), and according to different mutation patterns it was possible to differentiate histologies with an accuracy of 89.2% in the training set and 91.5% in the testing set.

However, one of the limitations of using ctDNA to identify histology is the extremely low detection rate in early stage adenocarcinoma, which may be around 13% [24,47]. For this reason, ctDNA detection in suspected lung cancer may be confirmatory for squamous-cell histology, while its absence could not permit the exclusion of adenocarcinoma, especially in part-solid nodules.

Regarding stage assessment, the association with computed tomography scans or PET information may ensure the best possibility to have precise stage assessment. It is quite clear from the literature that advanced stages are associated with higher ctDNA concentrations compared to early stages [23,24,46], with concentrations progressively increasing with stages.

Tran et al. [23] reported a ctDNA detection rate of 60% in a study involving 85 patients, with a detection rate of 49%, 58%, and 100% of patients with stage I, II, and III cancer. Similarly, Hong et al. [24] reported ctDNA presence in 31% of stage I and 71% of stage II patients using a large cohort of 895 patients, confirming a different detection rate according to the histology among different stages: 13% vs. 71% for stage I adenocarcinoma vs. stage I non-adenocarcinoma and 47% vs. 93% for stage II adenocarcinoma vs. stage II non-adenocarcinoma histology. The authors also noted that higher ctDNA levels were associated with TNM upstaging, suggesting that ctDNA information may guide surgical approaches and therapies, indicating extended resections and lymphadenectomy in these cases. Indeed, considering the risk of increased upstaging, lobectomy may be preferred to segmentectomy even in tumors smaller than 2 cm, with the possibility to perform a more extended lymphadenectomy in these cases. ctDNA may also correlate with recurrence risk and survival, suggesting that an aggressive surgical treatment may improve prognosis in patients with high ctDNA levels [24].

In summary, an increased ctDNA concentration, when revealed in adenocarcinoma, may be associated with stages more advanced than I, while it is difficult to associate stages with other histologies that usually present with a high detection rate. Similarly, when ctDNA is revealed in adenocarcinoma, there is a higher probability of predominant solid nodules and stages more advanced than I.

The possible association of ctDNA and other pathological features has not yet been deeply investigated, although some studies have investigated the association between ctDNA levels and mutation presence, especially in advanced stages, but there have been contrasting results. Lam and colleagues, in their study involving 144 advanced NSCLC patients, reported a significant correlation between ctDNA variant allele frequency (VAF) and tumor burden, evaluated according to RECIST criteria in CT volume (*p* ≤ 0.0001) and metabolic tumor volume (*p* = 0.003) terms. This correlation was strongest in KRAS and TP53 mutants and weakest in EGFR-mutated tumors. Conversely, Abbosh et al. [47] did not find any association between driver events in KRAS, EGFR, or TP53 and ctDNA detection.

On the other hand, the same authors found a significant correlation between ctDNA detection and necrosis, lymph node involvement, pathological tumor size, non-adenocarcinoma histology, lymphovascular invasion, and Ki67 labeling in univariable analyses, with the last three variables confirmed as independent factors in multivariable analysis. Hong et al. [24] reported that ctDNA detection was predictive of grade 3 tumors (*p* < 0.001) and PD-L1 expression (*p* < 0.001). These findings support the integration of ctDNA testing into routine diagnostic workflows in early stage NSCLC without the need for tumor tissue profiling. Furthermore, it is clinically useful in identifying high-risk patients who might benefit from innovative treatments, including neoadjuvant immune checkpoint inhibitors. (The results are summarized in Table 4).

## 4. Discussion

In the near future, ctDNA could represent a non-invasive, resource-sparing, and effective tool in clinical practice for screening programs in NSCLC, histology prediction, pathological features and staging, recognition of patients at a high risk for relapse, and identification of real progressions among pseudo-progressions of disease (Figure 1 and Figure 2).

The studies analyzed suggest that there may be room for the integration of ctDNA analysis into clinical workflows, particularly in early screening programs [45] and in settings requiring precision medicine approaches for patient stratification, identification of innate resistance, therapy optimization, and overall management [24,48,49,50]. An important ongoing study, the LANTERN project [51], aims to improve the prediction and stratification of prognosis for early stage lung cancer also through the analysis of ctDNA. These will help in forming a better characterization of the tumor so as to plan the best neoadjuvant and adjuvant treatment.

The implementation of ctDNA tests in clinical practice systems presents major challenges. While these studies advance the understanding of ctDNA’s clinical applicability, they also highlight the need for standardized methodologies and further research to address its limitations and expand its utility across broader patient populations. Indeed, by enabling non-invasive, dynamic monitoring of the therapeutic response or microscopic disease residual after treatments, relapse, or progression, ctDNA stands poised to redefine personalized medicine approaches to guide and facilitate timely and effective interventions. There are ongoing clinical trials (reported in Table 5) evaluating if resected stage I NSCLC could benefit from treatment intensification in the case of ctDNA positivity status after surgery and if ctDNA can predict recurrence for completely resected tumors.

The results of the studies discussed suggest that ctDNA could be a relevant diagnostic biomarker in patients with early stage NSCLC, both for diagnosis and for prediction of histology and stage. However, the clinical implementation of ctDNA testing of these patients remains far from being achieved. New fields of application are now the matter of ongoing clinical trials, which aim to combine this promising biomarker with new NSCLC therapies. The introduction of ctDNA into routine clinical practice needs standardization of ctDNA testing and appropriate prospective trials demonstrating the clinical value of incorporating ctDNA analysis in the management of patients [7].

## 5. Conclusions

In recent years, ctDNA testing is becoming an interesting option for the management of patients with early stage NSCLC. Emerging evidence suggests that the introduction of ctDNA in clinical practice may change the approach to the management of those with early stage lung cancer. Clinical trials with well-defined clinical objectives are needed to ensure that sufficient evidence is obtained to support the clinical adoption of ctDNA testing.

Future research should also focus on addressing current limitations, such as standardizing pre-analytical and analytical procedures and validating ctDNA assays in large-scale, prospective clinical trials. Ultimately, integrating ctDNA into multimodal diagnostic frameworks holds great potential for personalized therapeutic strategies, thereby improving outcomes for patients with early stage NSCLC.

## Figures and Tables

**Figure 1 diagnostics-15-00904-f001:**
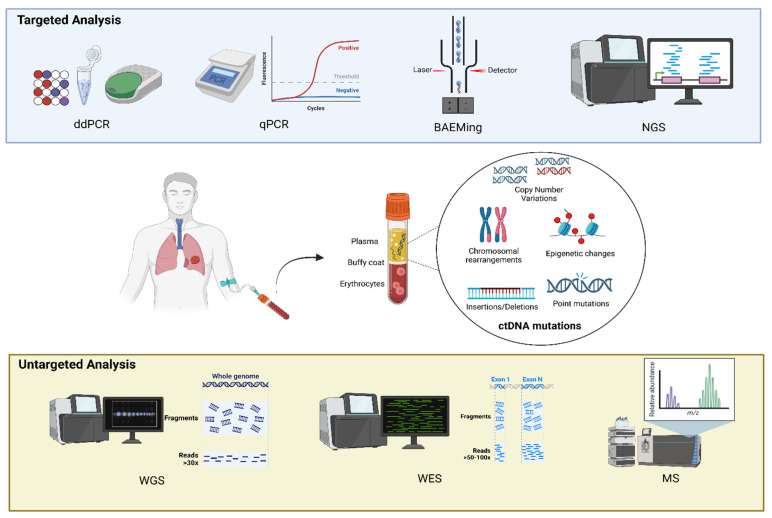
Kind of ctDNA mutation examined for tumor DNA detection and methods of analysis. Created with BioRender.com.

**Figure 2 diagnostics-15-00904-f002:**
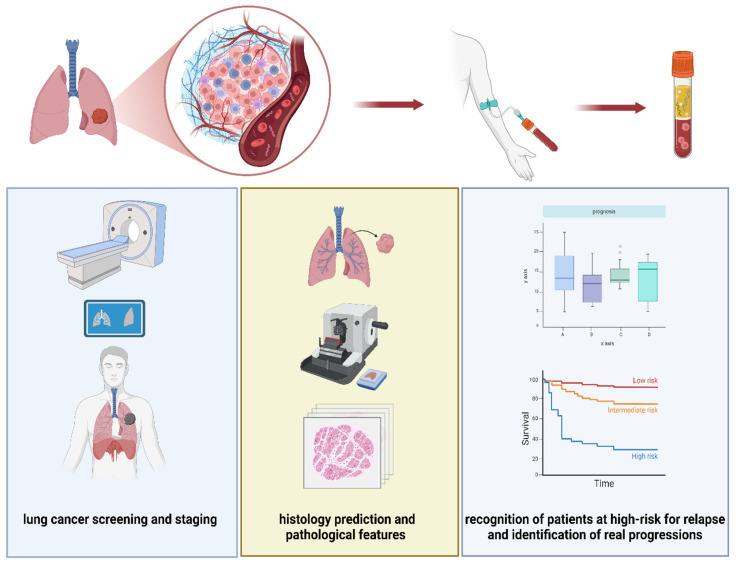
Possible application of ctDNA analysis in early stage lung cancer. Created with BioRender.com.

**Table 1 diagnostics-15-00904-t001:** Search strategy summary.

Item	Specification
Date of search	10 December 2024
Databases and other sources searched	PubMed
Search terms used	“ctDNA” and “early stage” and “NSCLC”
Time frame	1 January 2000 to 30 November 2024
Inclusion criteria	Original article, English language, and clinical trial (randomized, prospective, or retrospective)
Selection process	Two authors (C.S. and S.R.S.) independently reviewed the abstracts identified with this search, while a third author (F.L) was consulted in the case of discrepancies

**Table 2 diagnostics-15-00904-t002:** Methods of ctDNA identification summary.

Author	Period	Country	N of Patients	Histology and Stage	ctDNA Search Strategy	Analyzed Mutations on ctDNA
Santis [12]	2011	Austria	132	Adenocarcinoma (71.2%), squamous-cell carcinoma (12.8%), not specified (13.7%) Advanced stage	quantitative real-time polymerase chain reaction (qPCR)	EGFR, KRAS
Mosko [21]	2015	USA	122	Test performed on melanoma patients	UltraSEEK Oncogene Panel	BRAFV600E, EGFRG719S, KRASG13D, PIK3CA
Underhill [22]	2016	USA	N/A	N/A	Illumina sequencing	EGFR
Zviran [34]	2020	USA	N/A	N/A	ichorCNA test	circulating tumor allele fraction
Tran [23]	2023	USA	117	All NSCLC histologies Early stage	500 kb plasma-only assay prototype	ALK, BRAF, EGFR, ERBB2, KRAS, MET, NF1, NRAS, PIK3CA, PTEN RB1, STK11, TP53
Melton [32]	2023	USA	N/A	N/A	multicancer early detection (MCED) test	small-variant allele fraction
Lee [20]	2024	Korea	104	All NSCLC histologies Early stage	ULV1 panel and targeted next-generation sequencing (CT-ULTRA)	EGFR
Hong [24]	2024	Korea	414	Adenocarcinoma Stage I	methylation-based multicancer early detection (MCED) assay	EGFR, ALK
Bossè [25]	2024	Canada	260	All NSCLC histologies Stage I	Oncomine Precision Assay on the Ion Torrent Genexus System	EGFR, ALK, KRAS, BRAF, ROS1, MET, RET, NTRK, TP53, ERBB2
Rickles-Young [33]	2024	USA	34	N/A	Qiagen Circulating DNA Kit on the QIAsymphony liquid handling system	

**Table 3 diagnostics-15-00904-t003:** Baseline ctDNA at lung cancer diagnosis and screening summary.

Author	Period	Country	Kind of Study	Outcome in ctDNA Detection
Newman [18]	2014	USA	Prospective	100% of stage II–IV and 50% of stage I NSCLC patients
Chen [43]	2016	China	Prospective	NSCLC stage IA, IB, IIA: concordance of 50.4%, sensitivity of 53.8%, specificity of 47.3%
He [39]	2024	China	Prospective	AUC of 0.87 and an accuracy of 0.75 in detecting primary lung cancer
McKelvey [42]	2024	USA	Prospective	87.9% early stage lung cancer identified

**Table 4 diagnostics-15-00904-t004:** Baseline ctDNA rate histology prediction.

Author	Period	Country	Kind of Study	Outcome in ctDNA Detection
Chen [43]	2016	China	Prospective	GGO-predominant tumors had a cfDNA concentration more than 10-times lower that in solid tumors
Abbosh [47]	2017	UK	Prospective	No association between driver events in KRAS, EGFR, or TP53 and ctDNA detection
Zhang [46]	2019	China	Prospective	18% lung adenocarcinoma detection vs. 50% squamous-cell carcinoma detection
Hong [24]	2024	Korea	Prospective	22% lung adenocarcinoma identification vs. 81% squamous-cell carcinoma identification

**Table 5 diagnostics-15-00904-t005:** On going clinical trials on ctDNA in early stage lung cancer.

Clinical trial.gov Reference	Year of Registration	Kind of Study	Status	Primary Endpoint
NCT04585477	2021-04-08	Interventional	Recruiting	Change in ctDNA from trial enrollment to after two cycles of adjuvant durvalumab in stage I–III NSCLC patients who had a positive ctDNA status after surgery
NCT05921474	2023-04-03	Observational	Recruiting	To assess whether liquid biopsy for molecular residual disease during follow up can predict a recurrence of lung cancer
NCT06284317	2025-02	Interventional	Recruiting	To determine whether additional adjuvant immunotherapy with durvalumab after neoadjuvant chemoimmunotherapy has an effect on disease-free survival (DFS) in patients who do not achieve complete pathological response (pCR) as per local assessment according to the IASLC
NCT04712877	2022-06-15	Observational	Recruiting	Feasibility of comprehensive molecular profiling to detect actionable oncogenic drivers in patients with suspected early stage lung cancers scheduled to undergo biopsies to establish the diagnosis of lung cancer
NCT04638582	2022-08-28	Interventional	Recruiting	To predict the occurrence of a pCR based on the resolution of ctDNA detectability in early stage NSCLC
NCT03791034	2017-08-28	Observational	Recruiting	To evaluate whether peripheral circulating cell-free tumor DNA (cfDNA) can aid the screening of recurrence after complete resection of early stage non-small-cell lung cancer

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
