# Peer review of "The Role of ctDNA for Diagnosis and Histological Prediction in Early Stage Non-Small-Cell Lung Cancer: A Narrative Review"

_diagnostics, 2025, doi:10.3390/diagnostics15070904_

Round 1
Reviewer 1 Report
Comments and Suggestions for Authors
1. In Table 2, arrange all studies by year. In other tables too.
2. Line 353 mentions only one clinical trial, are there any others? If so, they should also be systematized in a table.
3. In general, ctDNA is used to determine mutations in genes. Was the correctness of the diagnosis in all the studies that were analyzed confirmed histologically?
4. Does it make sense to implement ctDNA technology if in all cases the tumor is removed, especially if we are talking about early stages? Especially since we are talking about fairly low accuracy.
5. For each study, add information on how many patients, with what stages of lung cancer, what histological type were examined. You can add it to Table 2.
Author Response
Reviewer 1
- In Table 2, arrange all studies by year. In other tables too.
Answer: thank you for this important observation. Studies are now arranged by year.
Correction: see all the tables
- Line 353 mentions only one clinical trial, are there any others? If so, they should also be systematized in a table.
Answer: thank you for this insightful comment. We went through the clinical trial now available on clinical trial.gov with the key words: early stage lung cancer, ctDNA and created a new table to summarize the main findings of the clinical trial active and recruiting to date.
Correction: see table 5.
- In general, ctDNA is used to determine mutations in genes. Was the correctness of the diagnosis in all the studies that were analyzed confirmed histologically?
Answer: thank you for this insightful comment. First of all, we assumed you related this comment to paragraphs 2 and 3, where we discussed about the possibility to predict if a lung nodule is a NSCLC through the analysis of ctDNA as a screening tool, and about the use of ctDNA to predict the histology. As you will find in table 3 and 4, we summarized the main findings of the selected studies. As you correctly asked, it is important to understand if the result obtained on the ctDNA analysis finds a confirmation on the surgical speciments. This would be the base of the implementation of the ctDNA in screening programs and in the NSCLC pre treatment work up also for early stage. Given these considerations, we found a good correspondence for ctDNA-lung cancer diagnosis, with some studies reporting a very high % of correct diagnosis [18, 42]. Concerning the prediction of the histology, a greater correspondence between ctDNA findings and histological sub type, has been found with the squamous cell carcinoma [24,26]. Once again thank you for the observation, as this helped us better analyse the correlation between ctDNA and histology.
Correction: see lines 254-257 and 314-317
- Does it make sense to implement ctDNA technology if in all cases the tumor is removed, especially if we are talking about early stages? Especially since we are talking about fairly low accuracy.
Answer: thank you for this interesting point of view, as it is very important to understand which could be the practical implication of the ctDNA implementation in early stages, for which the gold standard treatment is the surgical one, after which the analysis on the specimen can be performed.
In our opinion, the issue of the accuracy is an open one, indeed it is still under discussion which would be the best method to analyse the ctDNA. Concerning the identification of the ctDNA prior surgery, the advantage of the tumor-naïve approach provides a rationale for ctDNA-adaptive treatment escalation strategies before surgery, such as neoadjuvant immune checkpoint inhibitor therapy, in patients with high risk mutations. Of course this is all still matter of studies and ongoing clinical trials, so even thou this is a promising field, we are still far from a real implementation of the ctDNA for NSCLC at early stage. The aim of the review was to explore the status of the art of this topic and to understand which can been the potential role of the ctDNA in the setting of the precision medicine and surgery.
Correction: see table 5 and lines 339-342 and lines 347-349
- For each study, add information on how many patients, with what stages of lung cancer, what histological type were examined. You can add it to Table 2.
Answer: thank you for this important observation. We went through the selected work and implemented table 2 as suggested
Correction: see table 2
Reviewer 2 Report
Comments and Suggestions for Authors
The topic is not novel but applicable to the field and worth having review texts over it.
There are comment that should be corrected:
- the english litarture is not scientifically proper and should be improved.
- Illumination is a big weakness of paper. lack of impressive illumination. There are only two figures which are not informative and are very general. They should contain some specific information about ctDNA and previleges.
- A graphical abstract is very necessary for this kind of article.
- Keywords are not proper; for example, early stage is not a right keyword; rather, early stage diagnosis or small cell lung cancer are more informative.
- The aim of the manuscript is to analyze and offer an application for ctDNA as an early-stage marker. Some tables are also listed gene target and mutations indicating trials. but in discussion, it is not clarified, or there is no message which mutations or achievements could be applicable to translate these findings for readers to use these data. The authors should add and clarify such data to enrich the paper and improve its applicability of the paper.
There are some typos that should be modified, for example:
the line 356
In general the english literature should be improved
Author Response
Reviewer 2
The topic is not novel but applicable to the field and worth having review texts over it.
There are comment that should be corrected:
- the english litarture is not scientifically proper and should be improved.
Answer: thank you for this important observation. A mother tongue colleague, Dr Akshaya Balamurugan revised the whole manuscript.
Correction: through the text
- Illumination is a big weakness of paper. lack of impressive illumination. There are only two figures which are not informative and are very general. They should contain some specific information about ctDNA and previleges.
Answer: thank you for this important observation. The imagines have been revised and updated accordingly.
Correction: see figures 1,2,3
- A graphical abstract is very necessary for this kind of article.
Answer: thank you for your correct observation. The graphical abstract has been created
Correction: see graphical abstract
- Keywords are not proper; for example, early stage is not a right keyword; rather, early stage diagnosis or small cell lung cancer are more informative.
Answer: thank you for this important observation. Key words have been updated
Correction: see keywords section
- The aim of the manuscript is to analyze and offer an application for ctDNA as an early-stage marker. Some tables are also listed gene target and mutations indicating trials. but in discussion, it is not clarified, or there is no message which mutations or achievements could be applicable to translate these findings for readers to use these data. The authors should add and clarify such data to enrich the paper and improve its applicability of the paper.
Answer: first of all, thank you for this very interesting comment. As you correctly stated, the new technologies now in use let us identify gene targets and mutations also for the early stage lung cancer. By the way, for now, it is just experimental, as no target therapy has been approved yet for the early stages. In the near future, these findings will of course be included in the routine work up also for the early stage lung cancer to improve prognosis, and, where necessary, neoadjuvant and adjuvant therapy.
Correction: see lines 330-336 in the discussion section
Round 2
Reviewer 1 Report
Comments and Suggestions for Authors
I have no more comments on the manuscript.